# Using Scuba for In Situ Determination of Chlorophyll Distributions in Corals by Underwater Near Infrared Fluorescence Imaging

**Thomas Oh, Jittiwat Sermsripong and Barry W. Hicks \***

Department of Chemistry, United States Air Force Academy, USAFA, CO 80840, USA;
tomoh97@gmail.com (T.O.); C21Jittiwat.Sermsripong.th@edu.usafa.edu (J.S.)
**\*** Correspondence: barry.hicks@usafa.edu; Tel.: +1-719-333-6050

**Abstract:** Studies reporting quantitation and imaging of chlorophyll in corals using visible fluorescent emission in the red near 680 nm can suffer from competing emission from other red-emitting pigments. Here, we report a novel method of selectively imaging chlorophyll distributions in coral in situ using only the near infrared (NIR) fluorescence emission from chlorophyll. Commercially available equipment was assembled that allowed the sequential imaging of visible, visible-fluorescent, and NIR-fluorescent pigments on the same corals. The relative distributions of chlorophyll and fluorescent proteins (GFPs) were examined in numerous corals in the Caribbean Sea, the Egyptian Red Sea, the Indonesian Dampier Strait, and the Florida Keys. Below 2 m depth, solar induced NIR chlorophyll fluorescence can be imaged in daylight without external lighting, thus, it is much easier to do than visible fluorescence imaging done at night. The distributions of chlorophyll and GFPs are unique in every species examined, and while there are some tissues where both fluorophores are co-resident, often tissues are selectively enriched in only one of these fluorescent pigments. Although laboratory studies have clearly shown that GFPs can be photo-protective, their inability to prevent large scale bleaching events in situ may be due to their limited tissue distribution.

**Keywords:** Infrared Fluorescence Imaging; Chlorophyll; Fluorescent Protein; GFP; Coral Reef

## 1. Introduction

The reason phototrophs make chlorophyll is well known: photosynthesis. All phototrophs also have color and fluorescence due to the presence of chlorophylls and other photosynthetic pigments. In corals, the phototrophs are the endosymbiotic dinoflagellates *Symbiodinium sp.*, and each coral species preferentially harbors specific clades of *Symbiodinium* with different properties [1]. There are numerous Chl analogs in nature, each with unique spectral characteristics [2]. Although the relative composition of Chls in corals are influenced by the environment [3], the two most prominent Chls present in the known clades of zooxanthellae symbionts are chlorophyll a (Chla) and chlorophyll c2, with Chla being more abundant in most coral species. Chla has a complex absorption spectrum with prominent absorbance peaks in the blue at about 440 nm (the Soret band) and in the red at about 670 nm (Q band). Most Chls also fluoresce with complex excitation and emission spectra [4]. Chla has a peak fluorescence emission near 680 nm, but it also has a prominent shoulder in the near infrared (NIR) at 735 nm. The Chl emission from within photosynthetic organisms is not from free Chl, but instead from protein-Chl complexes in photosystem I (PSI), photosystem II (PSII), or the light harvesting complexes (LHCs) involved in photosynthesis [5]. Laboratory studies have shown a decrease in NIR fluorescence by extraction of the zooxanthellae pigments during bleaching events [6], but this has not been verified in situ. Examination of Chl fluorescence is valuable because along

with electron transfer and oxygen evolution, it can provide insight into the efficiency of PSI [7] and PSII [8], which are indicative of the overall level of photosynthesis occurring, and hence, the health of the zooxanthellae in the coral [4].

Most work attempting to image Chl fluorescence from corals has been done in laboratory aquaria, and most of it has focused on imaging the prominent Chla visible red emission peak near 680 nm [9–11]. Other coral pigments such as red variants of the green fluorescent proteins (GFPs) and cyanobacterial phycobiliproteins [12] can interfere with accurate Chl fluorescence analysis in that portion of the visible spectrum. It is surprising that few studies have attempted to image both pigments simultaneously, and again mostly in laboratory aquaria [13]. NIR imaging has been widely used to examine Chl fluorescence in the terrestrial environment [14], but its use underwater has lagged, because water absorbs NIR light 5–100-fold more efficiently than visible light, depending upon the wavelength [15]. To date, imaging zooxanthellae Chl fluorescence in situ requires either an imaging pulsed amplitude (PAM) fluorometer [16] or an underwater microscope [17], and although both of these studies utilized NIR illumination, neither study examined Chla fluorescence in the NIR. Only a few studies have imaged Chl with consumer cameras [18,19], and no attempt was made to separate Chl from other potentially interfering red fluorophores. Another commonly used method for direct in situ quantitation of coral Chl fluorescence is the diving PAM [20], which consists of an underwater fiber optic fluorometer placed directly onto the coral surface, but imaging with these instruments is limited to small areas. While the current methods are valuable for answering questions about individual polyps, they require unique and expensive instrumentation, personnel with extensive training and expertise to operate them, and none of these methods are capable of providing spatial information on a scale beyond perhaps the cm range. Imaging even a single modest coral colony would be difficult and massive colonies or whole reefs are beyond their capabilities.

In contrast to the relatively well understood role of chloropohylls, the reasons so many marine organisms including corals express GFPs remains unknown and controversial. GFPs have revolutionized biomedical research, and their discovery and development deservedly won the Nobel Prize in Chemistry in 2008 [21]. There are well over one million citations in the literature, and it is rare to pick up a top tier journal in life science or reef research today that does not contain at least one article utilizing or discussing GFPs. Entire books have been written on how to apply GFPs to solve complex research problems in the life sciences [22–24]. Though the first GFP identified was from the jellyfish *Aequorea victoria* [25], numerous color morphs including cyan, red, and yellow [26,27], as well as non-fluorescent chromoproteins [28], all with similar protein structures [29] have been isolated, cloned, spectrally characterized, and crystallized from scores of marine species. For simplicity, all the variants will be referred to here as GFPs. Although biofluorescence exists in both terrestrial and marine organisms [30], all of the known organisms possessing genes encoding for GFPs are from the marine environment, with most from phylum Cnidaria including jellyfish [31], corals [26,27,31], and anemones [32]. However, copepods [33] and even some marine cephalochordates have been found to express multiple GFP genes [34]. Additionally, marine fluorescence is not confined solely to GFPs [35]. A single organism can possess up to 16 genes encoding for different fluorescent and non-fluorescent GFP analogs [36], and whole genome sequencing of the coral *Acropora digitifera* identified at least 10 different GFP genes in that species [37]. Still, the functional roles for all these GFPs remains unclear [38,39], but it seems increasingly likely that the different isoforms may not have a single function. Instead, they may have evolved with multiple unique functions [38–41]. Proposed physiological roles for GFPs in corals alone include adaptive regulation of endosymbiosis [40], attraction of endosymbionts to juvenile corals [42], photo-protection of corals in shallow water [43,44], spectra adjustment in deep water corals [39,45], free radical scavenging in corals with excess illumination [46], protection from herbivorous fish [47], and at least a correlative role in immune response and rapid growth in coral under attack by predators or disease [48]. However, none of these roles appear to obligate, as non-fluorescent corals exist at all depths identical to species expressing GFPs [39], and different coral color morphs, due to differential expression of GFPs, in single species exist at a common depth [48].

Clearly, imaging zooxanthellae Chl fluorescence on a larger (colony or reef) scale could prove useful for scientific study and for conservation. Presented here is a novel method for providing direct images of Chl NIR fluorescence, from which zooxanthellae distribution can be inferred. Importantly, all of this work was done with commercially available equipment at modest expense in the natural environment, thus, it could be undertaken by the vast citizen scientist community of scuba divers. Chl NIR fluorescence can be imaged in daylight or at night, but when done at night it also permits comparative spatial analysis with GFPs.

## 2. Materials and Methods

Infrared Fluorescent Standards & Equipment. Chla was purchased from Sigma-Aldrich (St. Louis, MO, USA). Cameras, lenses, lighting, and underwater housings were from Backscatter (Monterey, CA, USA) and Blue Water Photo (Culver City, CA, USA).

Chl Excitation Lighting. For this work two 15,000 lumen VL15000P-Pro-MINI-TC lights from Bigblue were chosen (Clearwater, FL, USA). These lights have the advantage of including three different light emitting diode (LED) types, warm- and cool white as well as red LEDs. The lights were modified by caulking a 50 mm diameter, optical density 4, 675 nm short pass cut-off (CO) filter from Edmund Optics (Barrington, NJ, USA) over the source; most of this work was done with a 675 nm short pass filter (Edmund Optics #84-727), but other filters with 650, 700, and 750 nm COs were also examined for some work. This short pass 675 nm CO filter allowed maximum excitation, covering all of the major Chl excitation wavelengths including both the blue Soret excitation band and the red Q band, but stopped essentially all NIR emission from the excitation source.

Fluorescent protein lighting. Because red-shifted GFPs can interfere with Chl emission (especially the visible emission at 680 nm), and because the white excitation light possesses sufficient blue light to excite such GFPs, on night dives corals were first identified and imaged with two Light & Motion Sola Nightsea blue LED lights to select corals that were predominantly strong green emitters to the eye from 1–2 m. Prominent yellow or red emitters were not included in the study, except where noted in the text. This is also the reason why NIR imaging was chosen for quantifying Chl fluorescence rather than the more prominent 680 nm emission peak in the visible portion of the spectrum; the external 720 nm long pass NIR filter used for all chlorophyll imaging does not allow visible emission from other pigments to reach the camera sensor.

Camera Preparation and Use. The camera chosen for this work was the Panasonic GH5 mirrorless camera. Digital images from this camera are 20.2 megapixels. Some work was done using a Panasonic 14–45 mm f/4.5 Lumix G Vario zoom Lens, but the majority was done with a Panasonic Lumix G Vario 45 mm f/2.8 macro lens; both lenses have optical image stabilization. The close working distance of the macro lens ensured that NIR light was minimally absorbed by water after emission from the specimens. The camera has a 4/3 complementary metal oxide semiconductor (CMOS) sensor with highest sensitivity near 700 nm. The commercial camera comes with UV and IR filters and was converted to an infrared camera by Lifepixel (Mukilteo, WA, USA) for full spectral sensitivity. The underwater housing used for this work is the Nauticam (Seattle, WA, USA) NA-GH5 housing, with appropriate ports for each lens. In some of the in situ work, a +5 diopter from Sun & Sea was also added to the underwater housing port. Most imaging was done in P (program) mode to allow the camera to choose f/stop and shutter speeds, with the ISO typically fixed at 800 or 1600.

Filters and white balance settings. For normal visible images, no filters were used. For fluorescent images with blue light illumination, a combination of an ultraviolet/infrared (UV/IR) CO filter and a yellow filter were used in series. For NIR imaging a range of various external long pass IR filters (Neewar) were used; common cutoff wavelengths are 690, 720, 760, 850, and 950. For all the Chl fluorescent imaging a 720 nm long pass filter was used. Scattered excitation light has zero, or at most, minimal effect depending upon the excitation source LEDs and filter combination used; this was verified by direct imaging of the illuminating light sources, which showed very little signal, and then only with much slower shutter speeds than used in Chl imaging. The 67 mm ultraviolet/infrared (UV/IR) CO filter + yellow filter combination, and the NIR 720 mm external filters were inserted into a dual diopter holder attached to the camera port. When the camera is mounted on a tripod, this

configuration allows collection a series of nearly perfectly registered images that include, (1) a normal visible image with white light illumination, (2) a visible fluorescent (green, yellow, orange, and red emission) image with blue LED illumination, and (3) a NIR Chl fluorescent image with white light illumination. A digital image of the setup is shown in supplementary Figure S1.

A typical night sequence was begun by finding a sandy bottom, letting all the air out of the buoyance compensator and focusing on the white sandy bottom or a white dive slate to set three different custom white balances for each combination of illumination and filter. The three different custom settings were: (1) blue LED illumination with a UV/IR CO + yellow filter, (2) white LED illumination with no filters, and (3) white LED illumination with a 720 nm long pass IR filter. In some cases, a fourth custom setting with red LEDs and the 720 nm long pass filter was also used. Afterwards, the first custom white balance setting was selected, the blue LEDs lit, and the UV/IR CO + yellow filter moved into place over the camera port and the tripod moved into place near the selected coral. The camera drive mode dial was set to collect a series of three images, and the shutter was pressed three times to collect nine total images from which to choose. Each series of images were captured and saved in both RAW and JPEG format, in duplicate, on two ADATA 128 GB UHS-II memory cards. After the visible fluorescent images were captured, the blue LEDs were extinguished, the white LEDs lit, the custom white balanced changed to the second setting, the UV/IR CO + yellow filter removed, and a second set of three visible illumination images was collected. Finally, the third custom white balance setting was selected, the IR 720 nm long pass filter moved onto the port, and the final set of three images were captured. On most dives (~40 min) only corals within about a 10 m radius were imaged, thus, there were no significant changes in depth after the custom white balance settings were made. Since visible fluorescence is not remarkable during the daytime, daytime sequences included only visible images and NIR fluorescent images.

Scuba. The in situ images of coral Chl fluorescence were made in four locations: (1) the Main Jetty, Arborek Island, Raja Ampat, Indonesia with the assistance of staff from Barefoot Conservation in June of 2018 (typically 6–12 m depth with a 25 °C surface temperature); (2) Sandy bottom dive site at Carriacou, Grenada in January 2019 with assistance from Caribbean Reef Buddy and Deefer Diving (typically about 10 m depth with a surface temperature of a 26 °C); (3) the Egyptian Red Sea at the house reefs of Coraya Bay and Port Ghalib, Marsa Alam in February 2019 with the assistance of Coraya Divers (typically 12–24 m with a surface temperature of 22 °C); and (4) Key Largo, FL with the assistance of the staff from Coral Restoration Foundation (CRF) in March 2019 (typically 7–10 m depth with a 25 °C surface temperature). All dives were computer dives using 12 or 15 L of air. Night dives generally began within 15 min of sunset and ended in approximately 45 min. Nearly all work was done with the Nauticam housing mounted on a tripod and generally working as near to the minimum focal length (15 cm) that conditions would allow. For protection of the coral reef, this necessitated that only corals growing near a sandy bottom where the tripod could be safely mounted were imaged, and typically 2–4 kg of additional weight was carried to minimize current effects while on the bottom.

Image Analysis. For quantitative comparison of GFP and Chl distributions in various images, two methods were used. In the first, similar to [18], the visual fluorescence image was converted to an RGB stack to separate green and red channels to isolate the signal from the green GFPs. A single line plot profile was obtained using the Fiji version of NIH ImageJ software in both the green and red channels. A corresponding line was also constructed on the Chl NIR fluorescence image and pixel intensities were plotted as a function of distance from the same starting point. In the second method a 60 × 60 square region of interest (ROI) was placed onto the green channel image from the RGB stack to obtain the mean pixel intensity and standard deviation within the ROI. A corresponding ROI was also made on the Chl NIR fluorescence image at the same location and mean pixel intensity obtained in that ROI. The ROIs were moved to 40–50 different corresponding sites on the image pairs to evaluate the relative amounts of each pigment present.

Laboratory studies. UV-Vis and fluorescence spectroscopy was performed using a BioTek Synergy plate-reading spectrofluorometer (Winooski, VT, USA) with samples in 96-well UV-clear plates. All spectra shown are the average of triplicate samples and are corrected for sea water blanks.

Imaging chlorophyll a and coral skeletons was done with the same lighting and filter arrangements as the in situ work.

## 3. Results

### 3.1. Laboratory Spectroscopy

Figure 1a shows the absorbance spectrum of Chla, the most abundant Chl analog in most corals, in methanol. There are two prominent absorbance bands at 440 and 670 nm, although minor bands at 384 and 622 nm are also present. The strong absorbance in both the blue and red ends of the spectrum are responsible for the characteristic green color of Chla in solution. Figure 1b shows the fluorescence excitation and emission spectra for Chla. The excitation spectrum is essentially identical to the absorbance spectrum, indicating that there are no major absorbance bands without emission. The emission spectrum has a maximum in the red near 680 nm and a prominent shoulder near 735 nm. Given these spectra, it should be possible to selectively image only Chl fluorescence, perhaps quantitatively, above about 720 nm and this was verified in the laboratory (supplementary Figure S2).

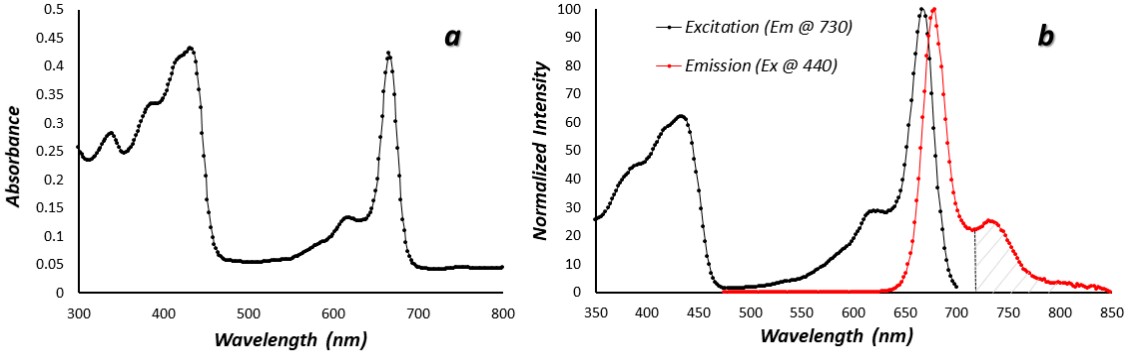

**Figure 1.** Chlorophyll a photophysical spectra in methanol solvent. (**a**) Simple absorbance spectrum in the near UV, visible, and NIR (**b**) Excitation (black) and emission (red) spectra of chlorophyll a, the main chlorophyll analog in corals. The cross-hatched region in the NIR above 720 nm shows the portion of the curve imaged in this work.

### 3.2. Control Imaging In Situ

Before attempting to image coral fluorescence, the method was validated using positive and negative in situ controls known to contain either large amount or zero Chl, respectively. All green algae possess relatively high Chla concentrations and *Halimeda* contains up to four times as much Chla as most corals [49], and since it is abundant, it is an excellent positive control. In contrast, most sponges do not harbor endosymbionts and would not be expected to show strong NIR fluorescence and could be good negative controls (although some sponge species imaged did show appreciable NIR fluorescent emission, this is consistent with published observations that sponges can contain endosymbionts [50]). Figure 2a,b show visible and NIR fluorescent images from *Halimeda* and Figure 2c,d show visible and NIR fluorescence images from a gray sponge obtained about 7 m depth in daylight. Several important factors emerge from these controls. First, sand or other particulates adhering to surfaces can prevent excitation light from reaching the corals as the dark spots on the top surfaces of the *Halimeda* (Figure 2b). Additionally, shadows cast from one portion of the subject to another in samples with complex three-dimensional (3D) structure (like all branching corals) can interfere with NIR fluorescence intensity when much or all of the illumination comes from nearby dive lights. Finally using a macro lens with the widest aperture (f/stop 2.8 with this macro lens) leads to a shallow depth of focus, and without collecting images at various distances and conducting extensive additional post image processing, this limits portions of the image in sharp focus.

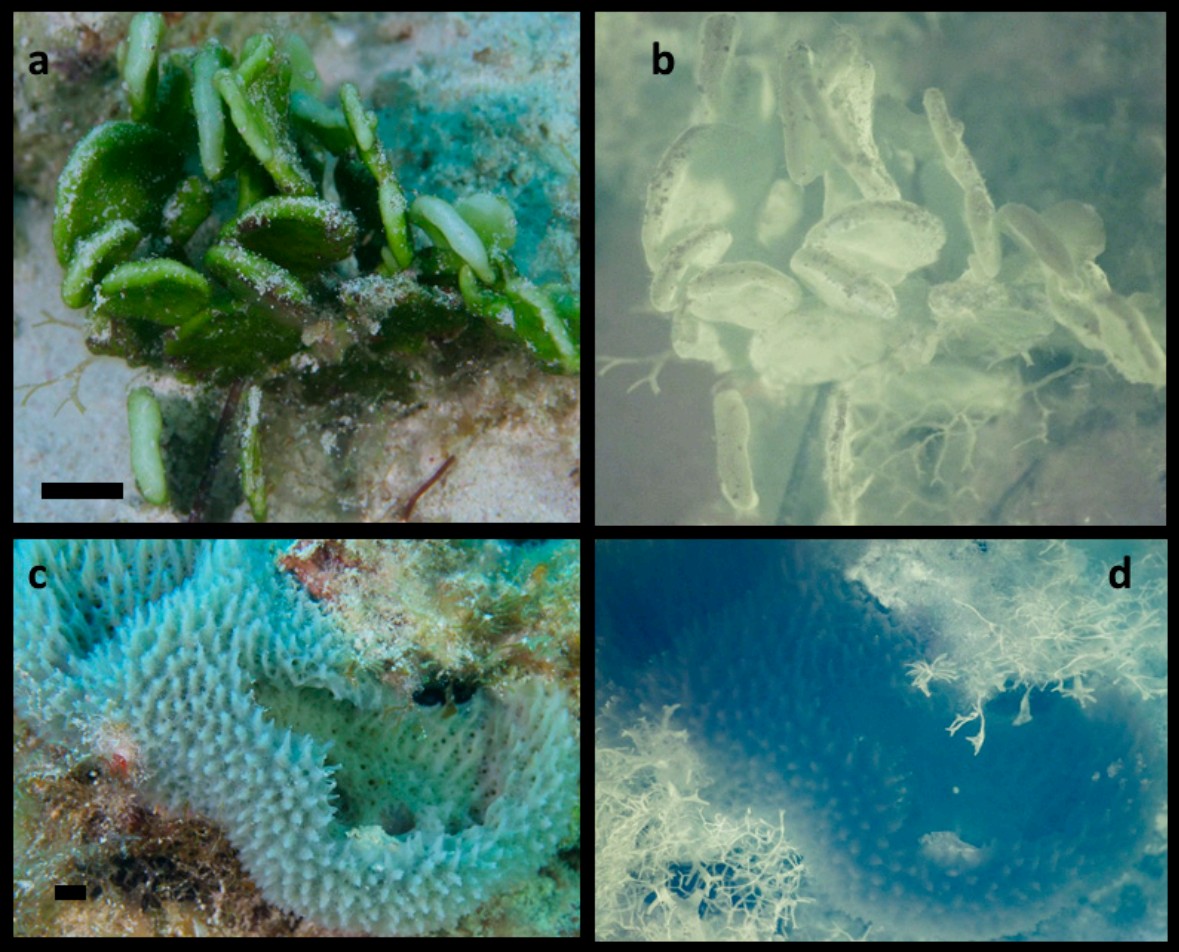

**Figure 2.** Positive and negative controls for NIR fluorescence imaging. (**a**) Visible image of green algae *Halimeda discoidea*. (**b**) NIR-image shows intense fluorescence from chlorophyll. (**c**) Non-photosynthetic sponge. (**d**) NIR-fluorescence image shows marine plants growing on the bottom over the edge of the sponge fluoresce, but the sponge does not. The ISO, f/stop are the same in b and d, but the shutter speed in d was twice as long. Scale bars in visible light images are approximately 10 mm.

### 3.3. Nighttime Imaging NIR Chl Fluorescence Imaging

Figure 3 shows the results of NIR Chl fluorescence imaging from a colony of *Montastraea cavernosa* in the Key Largo, FL at night. The tentacles are usually retracted in the daytime but extend shortly after sunset. In Figure 3a the visual image shows that most of the polyps are still in the act of extending tentacles. Figure 3b shows the distribution of Chl by NIR fluorescent imaging. In the center of the focal plane of this image there is abundant punctate fluorescence in the tentacles. However, there is generally much more intense emission seen from the mesentery. Furthermore, there is intense asymmetry in the mesentery Chl distribution due to fine structure in tissue that it is highly compartmentalized and also present in the visible images. Since all of the Chls are contained within the zooxanthellae, this image also shows where they are distributed in the coral tissues. Since Chl fluorescence is emitted in all directions, roughly half of the emitted light will travel toward the skeleton, and the coral skeletons have been shown to be highly reflective. Thus, our images will necessarily be a composite of those two sources of light, reflected and directly emitted, but both originate from Chl emission. Furthermore, to demonstrate that our imaging system at night does not produce enough NIR to be scattered/imaged, a control coral skeleton was imaged in the presence of a tube containing Chla in MeOH (Figure S4).

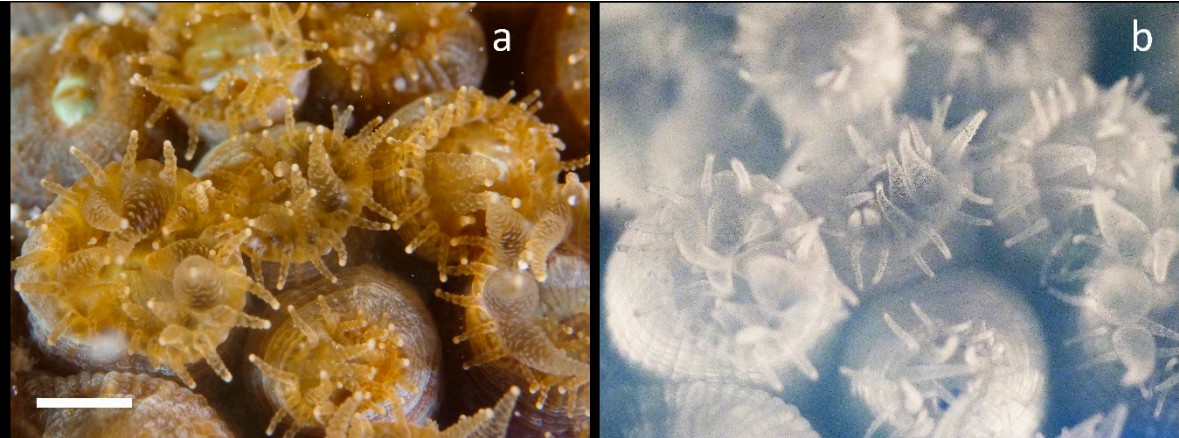

**Figure 3.** Nighttime in situ NIR imaging of chlorophyll fluorescence from *Montastraea cavernosa*. (**a**) Visible macro image shows that most polyps are partially extended. (**b**) NIR fluorescence image shows punctate fluorescence within some of the larger polyps, and brighter signals from the mesentery tissue with both radial and circular striations. The scale bar in the visible image is about 10 mm.

### 3.4. Daytime Imaging of Chl Distributions

Given the high absorption of NIR light in the water column it is not surprising that, in contrast to GFPs, which can only be imaged in daylight with complex imaging systems [17,18], it is quite easy to image NIR Chl fluorescence in the daytime. Figure 4 shows the results of daytime imaging in the Coral Restoration Foundation (CRF) nursery near Key Largo, FL. This is a green color morph of *Montastraea cavernosa* (Figure 4a). As expected for a near noontime dive, all of the tentacles are fully retracted and the oral disks and mouths are clearly visible. Figure 4b shows the NIR fluorescence from Chl in this colony. This colony was growing on the sea floor at about 7 m depth. Once again, compartmentalized fine structure in the mesentery tissue that is present both radially and circularly from the oral disk is seen. While the ambient scattered NIR radiation at this depth is miniscule [15,41], one cannot help but speculate that some NIR light scattered off the skeleton is present. It is noteworthy that the distribution seen in Figure 4b matches closely with that from the same coral species imaged at night (see below) when there is certainly no ambient NIR light to be scattered.

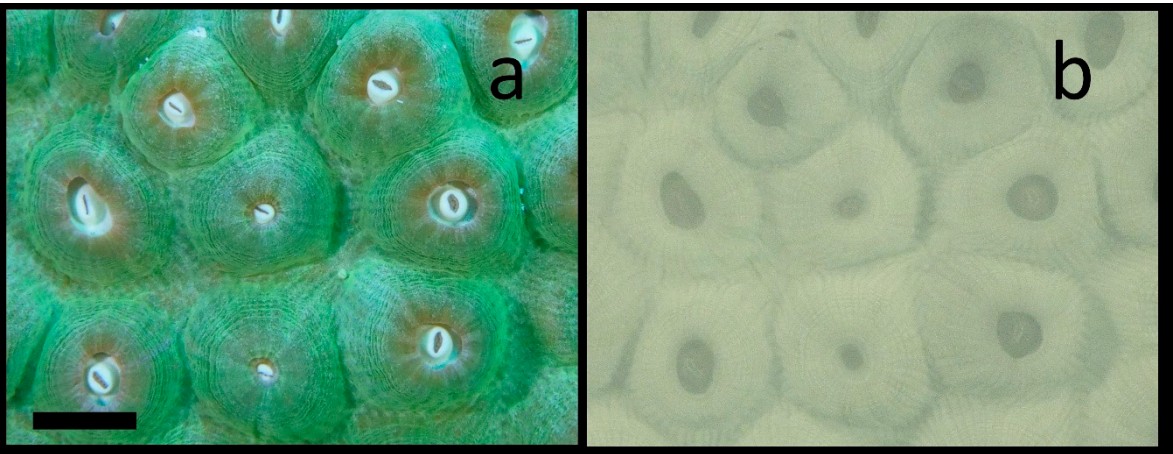

**Figure 4.** Daytime in situ NIR imaging of chlorophyll fluorescence from *Montastraea cavernosa* with external white LED illumination. (**a**) Visible image shows that most polyps are completely retracted in this green color morph. (**b**) NIR fluorescence image still shows clear asymmetric striations in each polyp. The scale bar in the visible image is about 10 mm.

It is noteworthy that the intense light sources were not required for daytime imaging of NIR Chl fluorescence, at least not at depths below about 4 m and as deep as 10 m. Figure 5 shows the images

of two coral species growing, while suspended from "PVC trees" in the CRF nursery at about 5 m depth. Figure 5a,b show the visible image and NIR fluorescence image with ambient daylight excitation, respectively, from a sample of *Acropora palmata*. Figure 5c,d are the visible and NIR fluorescence images with ambient daylight excitation, respectively, for samples of *Montastraea cavernosa*. Videos of the similar subjects indicate little or no scattered NIR light at this depth (evidenced as a lack of wave lensing that is readily detected at 1 m depth when imaging while snorkeling). The clarity of these images is not as good as previous images because neither the PVC tree, the camera, nor the diver were stationary; thus, each were subject to movements from the moderate currents at the time of imaging. Especially for the longer shutter times required for NIR fluorescence imaging, these movements make the images relatively blurry (compared to those captured with the camera on a tripod and the diver on the bottom). Still, in both of the NIR fluorescence images, individual polyps on the corals can be identified. NIR Chl fluorescence imaging with ambient excitation can be done at depths over 10 m, and even at distances of greater than 1 m from the specimen (supplementary Figure S3).

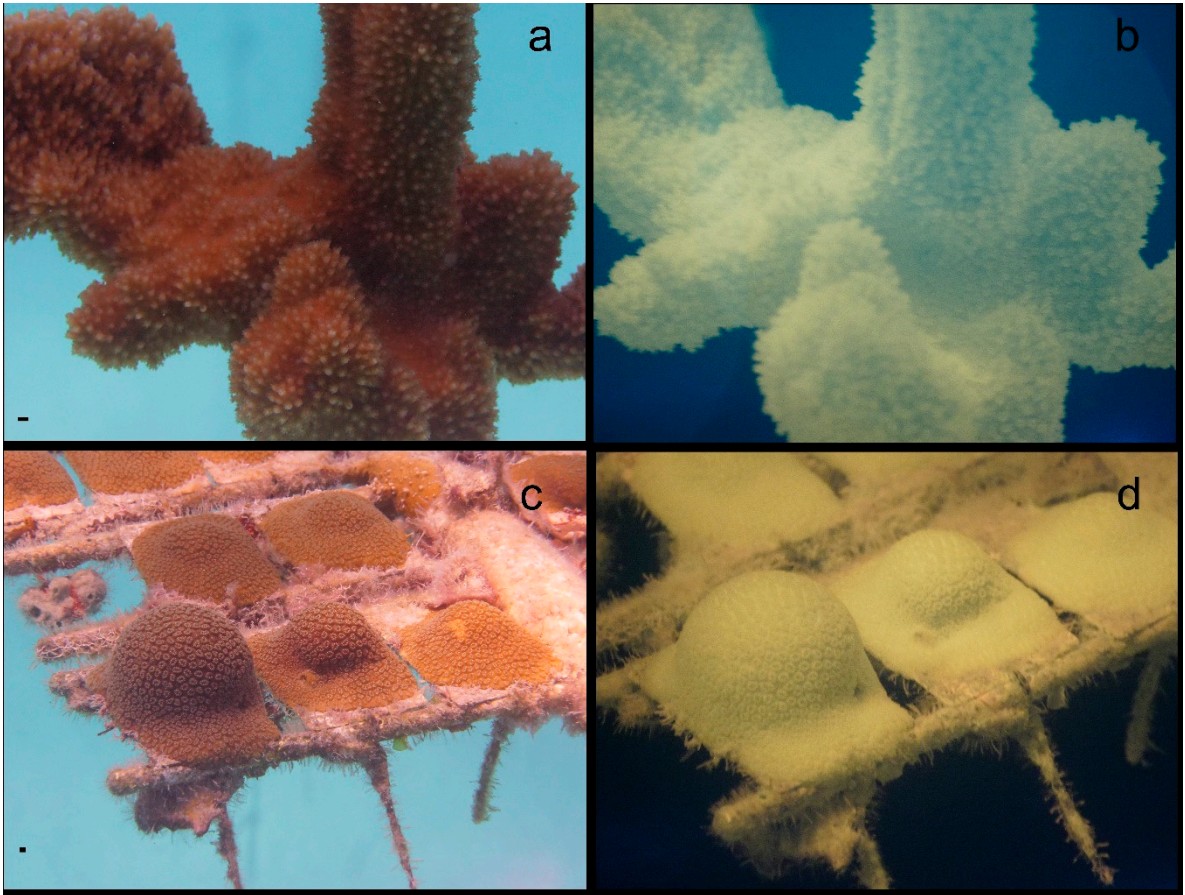

**Figure 5.** Visible and NIR Chl fluorescence images from two different Coral Restoration Foundation nursery coral species in ambient daylight. Images (**a**,**b**) are *Acropora palmata* suspended by a line on a "PVC nursery tree." Images (**c**,**d**) are *Montastraea cavernosa* growing on plates attached to a PVC tree. In the figure, (**a**,**c**) are visible images, (**b**,**d**) are NIR Chl fluorescence images. The scale bar in each visible image is approximately 10 mm.

### 3.5. Nighttime Imaging for Comparative Chl and GFP Distributions

To demonstrate the universality of this methodology and to examine the relative distribution of GFPs and Chl simultaneously, collections of visible, visible fluorescent and NIR fluorescent images were collected from corals in four worldwide locations over a 9-month period at various depths, temperatures, and times of day. Hundreds of corals were imaged, and thousands of images collected. A select few are shown in Figure 6 from three locations: (1) Carriacou, Grenada, (2) Marsa Alam in

Egyptian Red Sea, and (3) Key Largo, FL. All of these images were collected at night so that a comparison of GFP and Chl distributions could be made. Figure 6a–c are images of *Diploria strigosa*, Figure 6d–f are of *Acropora sp.*, and Figure 6g–i are of *Montastraea cavernosa*. Consistent with laboratory observations, most corals responded to blue light much more rapidly than they did to white or red LED illumination. Some species began rapidly retracting tentacles within 30 s when continuously illuminated with the blue LED video lights. Since blue LEDs were used to select corals expressing predominantly GFPs, as soon as corals were selected, the blue LEDs were shut off and one red LED source was used to illuminate the area during set up. In Figure 6a, extension of the tentacles is clearly seen in this coral, the tips of the tentacles are opaque relative to the majority of the tentacle shaft. The center portion of the ridges in this brain coral have very little visible pigmentation. The oral disks of a few polyps can be seen between the ridges. In Figure 6b, the vast majority of the GFP is located in regions where red fluorescence from Chl is not located, that is, on the central ridge and in the coenosarc between ridges. While some red fluorescence from Chl is seen in the tentacle tips corresponding to the opaque regions in 6a, by far the vast majority of red Chl fluorescence is seen in the mesentery tissues on the sides of the ridges co-localizing with the yellow pigmentation seen in the visual images, and it is virtually absent from the coenosarc. This drastically different, almost mutually exclusive distribution of GFP and Chl is confirmed in Figure 6c with NIR fluorescence. The center of the ridge has virtually no Chl, and most is in the tips of the tentacles or in the mesentery on the sides of the ridges. Co-localization of the visual yellow pigmentation with Chl is seen in the next coral as well. It is also noteworthy that the red fluorescence in the small fish on the surface of the coral in Figure 6b is not displayed in Figure 6c. In Figure 6d, the tentacles from many of the polyps of this *Acropora sp.* can be seen, but most remain retracted. The coenosarc on the theca shows the familiar striped yellow pattern characteristic of many *Acropora sp.* corals. In contrast the tops of the corallites near the polyp are relatively lacking in yellow pigmentation, especially near the actively growing tip of the coral. In Figure 6b, the GFP is distributed mostly on the theca and tops of the polyp, and enriched in the actively growing ends of the coral. Although there is more overlap in this species than the last, it appears to be most intense in regions where Chl and the yellow pigmentation are not present. In contrast, most the red fluorescence from Chl appears to be co-distributed within the yellow stripes seen in the visible images, as noted for the brain coral. This is confirmed in the NIR fluorescent images as seen in Figure 6e. Unlike the previous species, very little NIR fluorescence is seen within the tentacles. Additionally, in the actively growing tips where GFP expression is highest in the fluorescent images, Chl is all but absent. In Figure 6g, the tentacles of this coral are just beginning to emerge from some polyps. It is worth noting the sponge in front of the coral (which casts shadows from illuminating light sources), as well as the *Dictyota sp.* algae at the bottom of the image. If Figure 6h, GFP is much more broadly distributed in this coral, and is much brighter relative to the red Chl fluorescence seen in the other two samples. GFP is also seen enriched in the newly emerging tentacles, as is some red Chl fluorescence. The bright red fluorescence from the *Dictyota sp.* also acts as a positive control since these algae are rich in Chl. The sponges are generally not fluorescent, and in this case again acts as a negative control; the few red spots that can be seen are most likely from an adhering algae, and there appears to be a fine web of algal material over the end of the sponge in the visible image. Figure 6i shows distribution of photosynthetic zooxanthellae by the NIR fluorescence from the Chl. There is some punctate emission in the emerging tentacles, but it is much brighter in the mesentery tissue. Comparing Figure 6b,h, it is relatively difficult to see the distribution of zooxanthellae from the visual red fluorescence when GFP expression is higher, again providing value for the NIR fluorescence imaging methodology. Two difficulties in this imaging are noteworthy in Figure 6. First, the shallow depth of focus with wide apertures is apparent, and second, a shadow from the left light source that is not readily apparent in the visible fluorescence image, is clearly seen in Figure 6i.

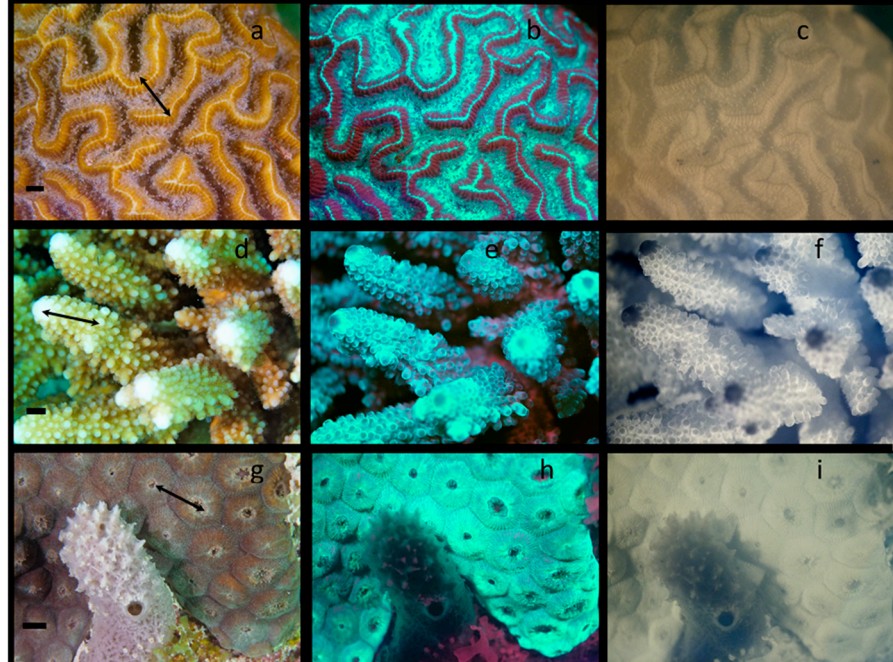

**Figure 6.** Images from three different coral species from different locations. Visible images (**a**,**d**,**g**); visible fluorescence images (**b**,**e**,**h**); and NIR Chl fluorescence images (**c**,**f**,**i**). (**a**–**c**) *Diploria strigosa* from Carriacou, Grenada in January; (**d**–**f**) *Acropora sp.* from the Egyptian Red Sea in February; (**g**–**i**) *Montastraea cavernosa* from the Florida Keys in March. Arrows on visible images represent the regions used for further digital image analysis in Figure 7. The scale bar in each visible image is approximately 10 mm.

A semi-quantitative analysis of the relative distributions of chlorophyll and GFP was undertaken from each pair of fluorescent images in Figure 6 (6b,c, 6e,f and 6h,i). That analysis is shown in Figure 7. First, the pixel intensities from GFP green emission, red emission (presumably mostly from Chl), and from Chl NIR emission were analyzed on a single line across each of the images (lines in Figure 6a,d,g). In Figure 7a–c, the relative intensities of GFP, red emission and Chl NIR emission are seen along a single line across the image. Although pixel intensities are not linearly correlated with pigment fluorescence (supplementary Figure S2), an increase in pixel intensity is indicative of increased concentration. Within any given coral sample there is a 3–4-fold difference in pixel intensity versus position on a polyp. Likewise, there is a 4–10-fold difference in pixel intensity for Chl NIR fluorescence. As noted above by visual inspection, it is clear from Figure 7a–c, that there are many tissues in every coral examined that indicate green GFPs and Chl diverge in nearly opposite directions (maxima for GFPs and minima for Chl, and vice versa). That there is little correlation between GFP and Chl distribution is verified in Figure 7d–f, where mean GFP and mean Chl pixel intensities within the same 60 × 60 square pixel ROI (roughly 2 × 2 mm) are compared. The extremely small values of the correlation coefficients indicate no relationship between GFP and Chl distributions. While the first two corals in Figure 6 have overall negative slopes in this analysis suggesting the possibility of reduced Chl fluorescence as GFP pigmentation increases, the opposite is seen for the last coral. Given the insignificant correlation coefficients, the large data scatter, and the lack of linearity in fluorescence imaging signals, this data should not be over interpreted but does provide an objective view of data in the Figure 6 images. Of note, the red channel fluorescence, while much more closely aligned with Chl NIR fluorescence in all three cases, does have some slight variation suggesting that part of that signal could come from red GFPs or other red-emitting pigments.

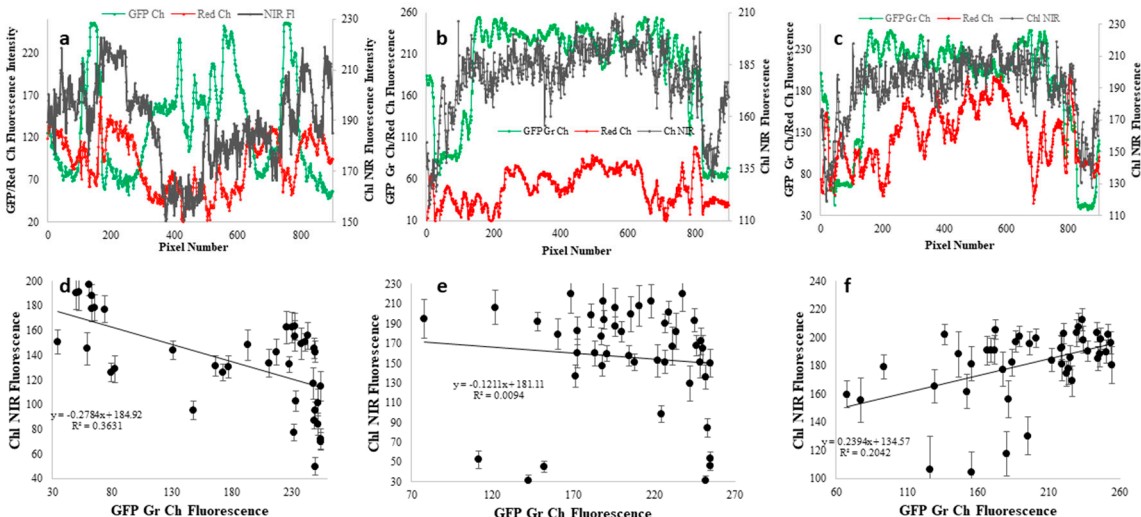

**Figure 7.** Digital image analysis of GFP and Chl distributions in corals from Figure 6. (**a**–**c**) The lines in Figure 6. represent the pixels regions used to determine the relative intensity of green GFP fluorescence (green curves), red fluorescence (from Chl, red GFPs and cyanobacterial pigments) (red curves), and Chl NIR fluorescence (gray curves) from the visible fluorescence (Figure 6b,e,h) and NIR fluorescence (Figure 6c,f,i) images. (**d**–**f**) Mean pixel intensities of green GFP fluorescence and NIR Chl fluorescence in 60 × 60 pixel ROIs randomly distributed throughout the images.

### 3.6. Resolution of Fluorescence from Red GFPs and Chl

To demonstrate the problem of using the visible red fluorescence as the sole signature of Chl, and the ability of NIR imaging to better identify Chl distributions in corals expressing large amounts of both green GFP and red GFPs, a specimen of *Siderastrea siderea* was found, that from a distance appeared to be bright green, but upon closer examination was apparently expressing both green and red GFPs. Figure 8a shows a magnified portion of a single *Siderastrea siderea* coral polyp. The tentacles are just beginning to emerge, and while green GFP is almost restricted to within the polyp cup, red GFP fluorescence from both the red GFP and Chl is clearly seen in the tentacles and, most intensely, in the coenosarc between individual polyps. Using digital image processing, the red and green channels of this image were separated, and the red channel only is shown in Figure 8b in 8-bit grayscale. The red fluorescence is brightest in the tentacles and in coenosarc near the borders of the individual polyps. In Figure 8c, the NIR Chl fluorescence image is shown. While the red GFP fluorescence overlaps with Chl on the tentacles, Chl is virtually absent from the region between the individual polyps where red GFP expression is high. While there is some co-localization of the GFPs and Chl, large portions of the coral tissues that have high Chl expression due to zooxanthellae, have little or no red or green GFPs present.

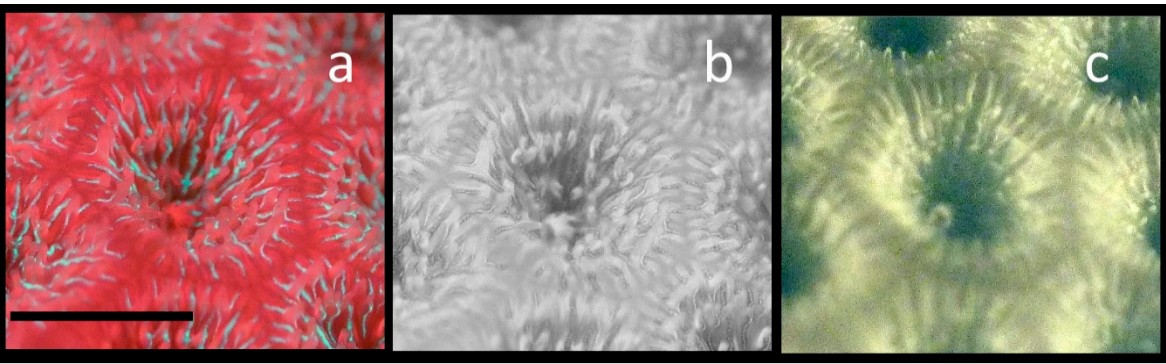

**Figure 8.** Demonstration of the ability to selectively image Chl even in the presence of red emitting GFPs. A single *Siderastrea siderea* polyp is shown. (**a**) Normal visible fluorescence image showing

green GFP and red GFP fluorescence along with Chl red fluorescence in a single polyp. (**b**) Red portion only of the RGB image from 9a. (**c**) The same polyp showing only the Chl NIR fluorescence image. For this image comparison, the contrast was enhanced in the NIR Chl fluorescence image. The scale bar in a is about 10 mm.

## 4. Discussion

One observation was obvious for most of the corals imaged in this work: in the native environment, green and red GFPs and Chla are not homogeneously, ubiquitously, or jointly expressed; many tissues where the zooxanthellae concentrations are greatest are virtually lacking in GFPs, and vice versa. Much of the in vitro work showing that GFPs have a photoprotective effect concentrates on non-fluorescent chromoproteins [44] and red GFPs [51], and indicates that the protection is most effective where those two GFPs absorb best, that is, in the yellow-orange portion of the spectrum. Furthermore, that in vitro work often uses "high and low" light fluxes for many hours, days, or even weeks to cause significant changes in GFP gene expression [43,44,52]. It has been proposed that the primary target of thermal and light induced coral bleaching is the antenna LHC of PSII in the endosymbiont [53]. While the in vitro work has unambiguously shown some degree of photoprotection for the coral under orange light illumination [43], to protect zooxanthellae from excessive light which can lead both to closing of PSII and production of reactive oxygen species (ROS) [54], the major portions of the spectrum that need to be covered are the blue (Soret band) and the red (Q band) where Chla absorbs most strongly. Since only cyan GFPs have decent coverage with the Soret band, and no GFPs cover the far red Q band, and since expression of cyan GFPs does not increase at all light levels [44], it may be that the laboratory observed phenomena are less important for photoprotection of corals in situ. We attempted imaging of Chl with all three LEDs: blue, red, and white. In all of the NIR images presented here, we used intense white light with a 675 nm short pass filter, because they were more intense and thus required shorter exposure times. This means we were simultaneously hitting both of major Chla excitation bands. Since most commercial blue and red LEDs have maxima near 440 and 630 nm, respectively, and white LEDs cover the entire excitation range it is also expected that white LEDs would produce the greatest fluorescence emission, and that was observed in controls. Furthermore, by exciting with light covering both major Chla bands (especially the major Q band), we could ensure that our NIR images would still show NIR Chl fluorescence, even when they were co-localized with, for example, cyan GFPs which would certainly lead to reduced Chl fluorescence if excitation was done using only the blue LEDs. However, when using white LEDs, which can also emit appreciable NIR light that could be reflected from targets, it is necessary to place a short pass NIR filter with a cut-off wavelength of 675 nm on the source to prevent the possibility of scattered excitation light interfering with the NIR fluorescence emission signal. Although some red-emitting GFPs have non-zero emission, even above 675 nm, and several cyanobacterial PBs have significant emission in the red, isolating the NIR emission above 720 nm is clearly more selective for monitoring only Chl distribution since no other common coral or symbiont pigments present in high concentration in the marine environment are known to emit significantly above 720 nm. Most importantly, while some coral species have shown resilience to both thermal-induced [55] and photo-induced bleaching [56], that large scale bleaching events occur increasingly often [57], from which few if any species of coral emerge intact, merely reinforces the need to follow up all in vitro work with in situ work.

To date, genetic transformation of zooxanthellae has proven to be very difficult [58], but when it is finally accomplished it will almost certainly be demonstrated with GFPs [59]. If GFPs can provide similar thermal- and photo-stability to *Symbiodium sp.* that it has been credited to provide to the corals [43,44], uptake of GFP-modified *Symbiodinium* by corals could be enough to generate a more photo-tolerant symbiotic coral-zooxanthellae duo capable of dealing with excessive illumination, especially with cyan GFPs which could protect from the more energetic blue excitation. Alternatively, if genetic manipulation of the corals [60] can be done to provide higher levels of ubiquitous GFP expression, that too might generate a more light-tolerant strain with greater resistance to bleaching.

In agreement with published ultrastructural work in anemone and black coral [61,62], in the focal plane of our best images there was abundant punctate Chl NIR fluorescence in the tentacles, and it is tempting to suggest this could be due to individual symbiosomes. However, there is generally much more intense emission seen from the mesentery tissue as expected [63]. Finally, there appears to be intense asymmetry in the Chl distribution in all corals due to fine structure in tissue that it is highly compartmentalized. In part, this is due to NIR Chl fluorescence emission scattered off underlying coral skeleton, as the coral skeletons are highly reflective [64]. Light emitted from excited fluorophores travels in all directions; thus, our images are a combination of light emitted directly from endosymbionts toward the camera sensor and light emitted toward, and reflected from, the coral skeleton. This is further complicated by the photon trapping, wave guiding and lateral light scattering within the coral tissue distances of up to 2 cm [65]. Additionally, many corals possess endolithic microbes within the skeleton that synthesize Chl, thus, some of the emission from some species in our work could be directly from such skeletal microbes [66].

Our work clearly shows that with a macro lens and intense full visible spectrum lighting, it is possible to image the polyp and Chl at night with spatial distributions nearly approaching that of the underwater microscope [17] and better than those using an imaging PAM [16] but at a fraction of the cost. We made numerous attempts to image internal standards with various Chl concentrations to try to bring a more quantitative aspect to our work, but that has proven to be fraught with difficulty and is still being developed. Yet, the relative intensities in the Chl NIR fluorescence images while not linearly quantitative, give at least a qualitative idea of Chl distributions in various coral tissues. Given that recent control work on PAM fluorometry, long considered the "gold standard" in Chl quantitation and imaging, has shown that the methodology has as much as 100% error in estimates of electron transport rates in corals in vitro [67], NIR Chl fluorescence imaging like we have done here should be further pursued and refined though it also does not have a linear relationship between intensity and concentration. Recent in vitro work using laser induced fluorescence to monitor the relative intensities of the 680 and 735 emission peaks for Chl fluorescence in corals [68] indicates that the ratio can be correlated with thermally induced bleaching. Modifying our NIR Chl fluorescence system to image this in situ would be quite simple.

Visual fluorescence has a remarkable effect on most people, and it is not surprising that the Professional Association of Divers Instructors (PADI) now offers fluorescent dives in its curriculum [69]. Since humans do not see in the infrared portion of the spectrum, digital imaging in this region has had less appeal to most divers. Anyone who is currently doing underwater photography can easily modify their cameras for NIR Chl fluorescence imaging, and without losing any flexibility. To return it to a pseudo-factory condition one only needs to utilize an external UV/IR cut-off filter. By adding a few filters for their lighting, divers can be imaging chlorophyll fluorescence while scuba diving. This means that NIR fluorescence imaging as we have introduced here has the potential to reach the vast "citizen scientist" community as well as full time researchers.

However, for scientists familiar with NIR fluorescence from Chl and its essential role as an indicator of photosynthetic performance, imaging in the NIR can provide a unique ability to selectively image chlorophyll without interference from other pigments. Moreover, while past research has suggested that cameras are not as effective as human vision in assessing the health of shallow reefs [70], that conclusion can only be accurate in the portion of the electromagnetic spectrum where humans see. Future work to improve this methodology is ongoing. First, the use of commercially available narrow bandpass filters near 735 nm is being incorporated to collect and correct for NIR emitted light reflected off the skeletons (735 fluorescence-735 reflectance) to improve selective identification Chl distributions. Second, we are initiating some ratio imaging studies (735 fluorescence-735 reflectance/700 fluorescence-700 reflectance) at 700 and 735 nm that might allow for direct quantification of Chl content in the corals [71]; although that work will require extensive post processing of data. Additionally, our data from an RGB camera must be compared to that from alternative imaging methods such as underwater hyperspectral imaging [72]. Finally, in situ work with both raw and corrected images at 680 and 735 nm [68] will be done to determine if this imaging methodology can be used to detect or identify onset of bleaching events; if so, application to

autonomous underwater vehicles could be useful in reef conservation efforts [73]. Despite the availability of numerous highly specialized techniques for obtaining specific information on corals for decades [16–18,74] no simple, inexpensive, accurate, broadly applicable method for monitoring coral reef health in the natural environment has been identified; clearly, this is a complex problem. In this initial work, we have demonstrated the potential for an inexpensive, commercially available, underwater NIR fluorescence imaging system to be applied broadly in situ for monitoring chlorophyll distributions.

**Supplementary Materials:** The following are available online at www.mdpi.com/xxx/s1, Figures S1–S4. These figures show 1) our imaging system, 2) that imaging Chl fluorescence can be quantifiable by comparison to spectroscopic data, 3) that in situ Chl imaging can be done at distances of up to 1 m from the specimen, and 4) that our white light illumination does not possess enough NIR light to be scattered off coral skeletons, proving the light emitted is from Chl.

**Author Contributions:** All three authors collected data while Scuba diving. Both T.O. and B.W.H. collected laboratory data. All three authors contributed to the writing and editing of the manuscript, although most of the editing during and after the initial submission was done by the PI and corresponding author. Conceptualization, B.W.H.; Methodology, B.W.H.; Formal Analysis, T.O. and B.W.H.; Investigation, T.O., J.S., and B.W.H.; Resources, B.W.H.; Data Curation, B.W.H.; Writing—Original Draft Preparation, T.O. and B.W.H.; Writing—Review and Editing, T.O., J.S., and B.W.H.; Supervision, B.W.H.; Funding Acquisition, B.W.H. All authors have read and agreed to the published version of the manuscript.

**Funding:** We would like to thank the US Air Force Office of Scientific Research and the Department of Chemistry and the Chemistry Research Center at the US Air Force Academy for funding this project.

**Acknowledgments:** We would also like to thank the following individuals organizations for assistance with scuba diving: Amelia Muora and staff at the Coral Restoration Foundation in Key Largo, FL; Matt Readout and Gary Ward with Caribbean Reef Buddy and Deefer Diving in Carriacou, Grenada; Kyle Hunter and the Barefoot Conservation Staff on Arborek Island, Raja Ampat, Indonesia; Coraya Bay Divers and the JP Marine Liveaboard staff in Marsa Alam, Egypt. Special thanks to Cynthia Corley for assistance for purchasing equipment.

**Conflicts of Interest:** The authors declare no conflict of interest.

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
