# Peer review of "Using Scuba for In Situ Determination of Chlorophyll Distributions in Corals by Underwater Near Infrared Fluorescence Imaging"

_jmse, doi:10.3390/jmse8010053_

Round 1

Reviewer 1 Report

I read this article with interest. The authors have done a nice study to advocate that an amateur diving community could equip their underwater cameras for modest costs to measure Chl in situ w/o interference from the GFPs.  I have two negative comments about this article that make it hard for me to endorse its being published in its current form:

1) The claim that all light seen in the low pass images is due to fluorescence.  Given the fact that coral skeletons are white (vis-a-vis) very reflective, I dispute the claims that the white striations in Figure 4b that consist of the radial streaks are due to Chl.  This may well be to just a reflective skeleton in the image that is reflective enough in the ambient NIR. Surely, some of the other images are quite nice and one can even see some subsurface particulates in Figure 3b that seem to be from the symbionts.  However, it's hard for me to imagine that all of the light seen in the NIR is due to Chl.. as this is one counter-example. 

2) That this article provides a nice, inexpensive venue for citizen science.  Treibitz et al. provide a nice article that can easily be adapted by a user community to look at the fluorescence from corals.   Without going into the details of that article, if the primary goal of this one is citizen science, it seems like that one is already playing such a role.  Note that they have extensive group truth via in situ spectrometry that is missing here.  RGB cameras have wide band sensors for each of the channels.  This does not preclude the type of analysis here but their work seems to have the kind of ground truth that warrants publication.  

In a more general sense, it's a bit out of the ordinary for a scientific journal such as JMSE to publish articles whose main goal is to provide a vehicle for citizen science.  Although I didn't dismiss this on a fundamental basis, it does seem to me that the "science" would need to be a bit better to being published in JMSE.  Perhaps Marine Technology would be a better venue.  It's actually more popular among ocean scientists and equipment manufacturers. 

Author Response

Responses to 1st Reviewer:

1) The claim that all light seen in the low pass images is due to fluorescence.  Given the fact that coral skeletons are white (vis-a-vis) very reflective, I dispute the claims that the white striations in Figure 4b that consist of the radial streaks are due to Chl.  This may well be to just a reflective skeleton in the image that is reflective enough in the ambient NIR. Surely, some of the other images are quite nice and one can even see some subsurface particulates in Figure 3b that seem to be from the symbionts.  However, it's hard for me to imagine that all of the light seen in the NIR is due to Chl. as this is one counter-example. 

There are several issues raised here.  Let me respond to the reviewer’s statement that “I dispute the claims that the white striations in Fig 4b that consist of radial streaks are due to Chl.”

Our white light sources used for this work have an optical density 4.0 675 nm cutoff filter over them.  The camera has a 720 nm long pass filter over the lens.  There is very little NIR light above 720 being emitted by the lights (direct imaging of the lights themselves with the camera using the same settings used in this work shows the lights to be detectable, but not bright).  Thus, if the reviewer is arguing that our images are merely NIR light from our sources reflected off the coral skeleton it is inaccurate as there is simply not enough NIR light from our light sources to begin with.  To demonstrate this, we have added an additional supplementary figure (new Figure S4) to show this.  There is so little NIR light reflected off coral skeletons relative to that emitted by Chl as to be essentially negligible.

If the reviewer is arguing that solar NIR light is the main contributor to the NIR fluorescent images, after being reflected off of the coral skeleton, this also is unlikely.  The image for Fig 4b was collected at a depth of about 7 m (24 ft).  Visible red light at 600 nm penetrates sea water to a depth of about 5 m, NIR light above 700 nm has a penetration about 5-50 times less than red light  (see 1) Eyal, Gal, et al. "Spectral diversity and regulation of coral fluorescence in a mesophotic reef habitat in the Red Sea." PloS one 10.6 (2015): e0128697 and also 2) Pegau WS, Gray D, Zaneveld JRV (1997) Absorption and attenuation of visible and near-infrared light in water: dependence on temperature and salinity. Applied optics36(24), 6035-6046.).  While it is certainly true that there is a minute, non-zero amount of NIR light at 7-8 m, it is very small.  Furthermore, and most importantly, the same coral species imaged after sunset using our methodology with the external lighting described above (Fig 6i) shows essentially the same pattern, although polyps are beginning to emerge in that image.  Under those conditions, the “ambient NIR” this reviewer mentions is certainly next to zero.  Also for the record, using our imaging system while snorkeling in the daytime and collecting videos at 1-2 m depth, clearly shows a great deal of scattered/reflected solar NIR light, even off of soft coral surfaces with no skeletons and this is easily seen as wave lensing in videos.  In shallow water dives, even at about 5 m, wave lensing of solar NIR light is no longer seen, even though wave lensing of visible light is still seen.  Lastly, similar NIR fluorescence images can be captured from coral surfaces even at depths exceeding 15 m (54 feet was the deepest where we attempted ambient daylight excitation, although with longer exposures), where there is certainly next to zero ambient solar NIR light interfering; that must certainly be from blue light excitation of Chl.

However, the point is well taken that after illumination, light is emitted from Chl in all directions.  Roughly half of the emitted light will certainly travel toward the coral skeleton, and reflect off of that skeleton, before coming to the camera sensor.  A portion of the other half will travel directly toward the sensor.  The NIR fluorescent images we have are a combination of directly emitted and reflected NIR light from excited chlorophyll.  But make no mistake about it, all of the NIR light imaged here is emitted from chlorophyll. This reflection of emitted light is also seen in Figure S4.  Compare in that figure the intensity of the small coral specs directly below the tube of Chl and from those that are not beneath that tube.  In places where the concentration of chlorophyll is high, and the coral tissue thin (or the skeleton closer to the tissue surface), it will appear brighter and take on some of the skeletal pattern…but that is still because there is chlorophyll present in that area.  Reflection of emitted NIR light from Chl off of the coral skeleton where there is almost no Chl present to begin with, would certainly not account for the spatial distributions we documented here (compare the mesophyll tissue in Figs 4b and 6i with the tissue near the coral mouths).

Finally, the coral skeletons themselves can possess Chl (see Ralph, Peter J., Anthony WD Larkum, and Michael Kühl. "Photobiology of endolithic microorganisms in living coral skeletons: 1. Pigmentation, spectral reflectance and variable chlorophyll fluorescence analysis of endoliths in the massive corals Cyphastrea serailia, Porites lutea and Goniastrea australensis." Marine Biology 152.2 (2007): 395-404.); so again, some of the emission imaged could be directly from the coral skeleton; varying distributions of endolithic microbes producing Chl to different extents could contribute to differences observed among different corals but that is an issue beyond the scope of this work.

None of this however alters our conclusion: while light traveling through coral tissue is complex (see Wangpraseurt, Daniel, et al. "Lateral light transfer ensures efficient resource distribution in symbiont-bearing corals." Journal of Experimental Biology 217.4 (2014): 489-498.), the light we are imaging is all being emitted from chlorophyll (unless there are as yet unknown fluorophores that emit in the same portion of NIR in these organisms). To address this point of the reviewer, we have added supplementary figure 4 of a coral skeleton, and a 3/8 inch section of that skeleton in a 4 L beaker of water, along with a dilute solution of Chla in MeOH (thus, imaged through about 25 cm of water comparable to what we used for most of our imaging, with camera settings similar to those from our in situ work) to show that there is almost zero NIR light reflecting from the skeleton with our lighting/imaging system under conditions where Chl is easily detected.  We have also altered the text to accurately include this description, and included the missing references listed above. 

2) That this article provides a nice, inexpensive venue for citizen science.  Treibitz et al. provide a nice article that can easily be adapted by a user community to look at the fluorescence from corals.   Without going into the details of that article, if the primary goal of this one is citizen science, it seems like that one is already playing such a role.  Note that they have extensive group truth via in situ spectrometry that is missing here.  RGB cameras have wide band sensors for each of the channels.  This does not preclude the type of analysis here but their work seems to have the kind of ground truth that warrants publication.  

The reviewer is correct that we should have referenced this work from Treibitz et al. (we did reference other work from this group, and the text has been altered to include this reference).  However, while Treibitz et al. did use a commercial camera, and they did image both fluorescent proteins and chlorophyll, all of their images are predominantly from blue light illumination, and the vast majority of their Chl emission is in the red near 680 nm; they made no attempt to separate visible and NIR emission from chlorophyll.  As mentioned in our paper, there are several known fluorophores that could interfere by emitting red light and thus, complicate the interpretation when red fluorescence is attributed solely to Chl; this is much less likely in the NIR.  The fact that that work has 23 citations in just a few years, also indicates that pursuit of work that can be adapted by citizen scientists is of value, for any scholarly source; scientists don't "own" science.  Also, in places where GFP and Chl are co-localized, the GFPs will absorb much of the blue light and could lead to apparent reduced Chl fluorescence and make it much more difficult to quantify with 680 red emisison.  Finally, as pointed out by this reviewer, RGB cameras like the Cannon camera used in that work and the Panasonic used in our work generally have a Bayer filter pattern, with 50% green pixels and only 25% each in the red and blue.  Thus, imaging both green and red fluorescence and simply separating the channels is fraught with complications, requiring more extensive characterization.  All of the pixels, green, red and blue, respond to NIR light, albeit with non-identical efficiencies.  I don’t know if the reviewer saw our supplementary figures or not, but we did do some spectrofluorometry.  If one compares figures S3d and S3e, it is clearly shown that both imaging and fluorometry have comparable curves with increasing emission signal as Chl concentration increases.  The fluorometry has a wider linear (dynamic) range, but that would be expected as the PMT tube is designed with that in mind.

In a more general sense, it's a bit out of the ordinary for a scientific journal such as JMSE to publish articles whose main goal is to provide a vehicle for citizen science.  Although I didn't dismiss this on a fundamental basis, it does seem to me that the "science" would need to be a bit better to being published in JMSE.  Perhaps Marine Technology would be a better venue.  It's actually more popular among ocean scientists and equipment manufacturers. 

I would not have targeted this work to this journal until I saw the special issue on Underwater Imaging.  There is very little work published on underwater NIR imaging.  There is virtually no work published on underwater NIR fluorescence imaging.  I would imagine the editor of an issue dedicated to underwater imaging would like to see highly varied contributions bringing new technology to the field, but that is why I chose to submit this work first to this journal; without such a special issue I would have pursued other journals.

Reviewer 2 Report

The paper by Oh et al. is interesting and presenting an old method used in remote sensing of terrestrial environments with an adaptation to the aquatic world. The importance of Chl quantification in-situ over a large scale is tremendous, unfortunately, your method is not able to do it. But, the paper has some important data and methodologies that should be acknowledged and published for the future development of a quantitative method or calibration.

Specifically, I have a few comments on the original text that should be addressed before publication.   

Intro:

Lines 28-31: need to revise according to new findings (see: Systematic revision of Symbiodiniaceae highlights the antiquity and diversity of coral endosymbionts), in addition, ref [1] is not appropriate!

Lines 51-53: Why do you clam that there is a correlation between GFPs expression and Chl content? The GFPs could be in several layers of the coral tissue, see for example Ben-Zvi et al. 2015: Light-dependent fluorescence in the coral Galaxea fascicularis.

Line 89: see more about mesophotic coral’s fluorescence in Eyal et al. 2015 (PloS ONE) and Ben-Zvi et al. 2019 (Scientific Reports)

M&M:

Line 184: typo “first”

Line 194: (Winooski, VT)

Lines 195-196: if it is “average of triplicate samples” were are the error bars or confidence interval levels of each point? Please correct the spectra figs.

Results:

Line 285: panel c and d are in different magnification (also a & b but it is less notable), please correct both of the pairs.

Lines 300-302: I guess you have a component of UV in your blue led see Ben-Zvi et al. 2019 - Response of fluorescence morphs of the mesophotic coral Euphyllia paradivisa to ultra-violet radiation.

Line 345: What the arrows at panels a, d and g pointing? You should have a description in the caption…

Lines 364-366: but in the intro lines 51-53 you claimed differently. Should be corrected in the intro…

Line 401: 8a and fix the panel to put c closer to b as a is…

Cheers and good luck,

Gal

Author Response

Response to 2nd Reviewer’s Comments:

We greatly appreciate this reviewer’s constructive criticisms.  In every instance but one, we have altered the text exactly as suggested by this reviewer.

Intro:

Lines 28-31: need to revise according to new findings (see: Systematic revision of Symbiodiniaceae highlights the antiquity and diversity of coral endosymbionts), in addition, ref [1] is not appropriate!

This change was made exactly as suggested.

Lines 51-53: Why do you clam that there is a correlation between GFPs expression and Chl content? The GFPs could be in several layers of the coral tissue, see for example Ben-Zvi et al. 2015: Light-dependent fluorescence in the coral Galaxea fascicularis.

The text was changed exactly as indicated (and also because it agrees more with our findings) to remove this statement.  This isn’t really our claim, but there have been studies suggesting increases in FPs could be used to “diagnosis” bleaching due to heat stress that is concurrent with increased expression of GFPs (Smith-Keune, C., and S. Dove. "Gene expression of a green fluorescent protein homolog as a host-specific biomarker of heat stress within a reef-building coral." Marine Biotechnology10.2 (2008): 166-180.).

Line 89: see more about mesophotic coral’s fluorescence in Eyal et al. 2015 (PloS ONE) and Ben-Zvi et al. 2019 (Scientific Reports)

We have added the Eyal reference to our manuscript and cited it in two places exactly as suggested.  While we agree that the Ben-Zvi work adds additional information, since it concentrated mostly on UV illumination, we did not include it.

Line 184: typo “first”

This was fixed exactly as suggested.

Line 194: (Winooski, VT)

This was added to identify source of BioTek spectrofluorimeter exactly as suggested.

Lines 195-196: if it is “average of triplicate samples” were are the error bars or confidence interval levels of each point? Please correct the spectra figs.

At this point the first author has graduated and is in Medical School.  The second author and I have the data, but we are beginning a new semester, and are very busy.  If this figure was in the body of the paper, and not a supplementary figure I would make the requested change.  But, I really don’t feel that it alters any conclusions of the paper, and doing hours of “busy work” that makes no substantial difference in our findings is just that, “busy work’; we did mention in the legend that triplicate variation was less than 10% in all cases.

Line 285: panel c and d are in different magnification (also a & b but it is less notable), please correct both of the pairs.

This was due to cropping these images to make it clearer for the reader, but this figure has been updated with the same cropping/magnification exactly as suggested.

Lines 300-302: I guess you have a component of UV in your blue led see Ben-Zvi et al. 2019 - Response of fluorescence morphs of the mesophotic coral Euphyllia paradivisa to ultra-violet radiation.

I have not seen the spectral output of the Blue LEDs, but I have requested it from the manufacturer.  This is certainly possible, and of note for anyone imaging coral fluorescence in situ since longer illumination times could negatively impact the animals.  I don’t think this is covered in the PADI course!

Line 345: What the arrows at panels a, d and g pointing? You should have a description in the caption…

The legend to this figure has been modified exactly as suggested by this reviewer.

Lines 364-366: but in the intro lines 51-53 you claimed differently. Should be corrected in the intro…

This was fixed in the intro exactly as suggested by this reviewer.

Line 401: 8a and fix the panel to put c closer to b as a is…

This was fixed exactly as suggested by this reviewer.

Round 2

Reviewer 1 Report

After reading the new additions I'm ok with this being published now. i think that there are a few modifications that i would like the authors to consider:

1) There are a few self-laudatory expressions that I would encourage them to remove.  In general, they don't really add content and are a bit self proclaiming.  For example:  "Given that virtually all of the Chl in corals is contained within the zooxanthellae, our NIR Chl fluorescence images are  among the best and most selective images of zooxanthellae distributions in coral in situ ever obtained".

2) The authors rightfully argued that the depth of the corals would preclude the NIR light from penetrating that far down. However, this new version does highlight that NIR emission due to CHL fluorescence could be reflected from skeletons and would then be part of the images. I'm not sure that we would say "mistaken' but that would be my comment.  This is the major improvement that adds content that warrants publication.

3) In general, it's always nice to point out some future directions with provisos on the limitations of the methods. So.. for example they do agree that (quote from them: "Thus, imaging both green and red fluorescence and simply separating the channels is fraught with complications, requiring more extensive characterization. All of the pixels, green, red and blue, respond to NIR light, albeit with non-identical efficiencies."  I think it would be nice to include such a statement in the text.  In addition, it seems like more advanced narrow band studies that looked at multiple narrow band reflections with concomittent imaging technologies (narrow band) could resolve the conundrum of definitely assigning reflected NIR vs inherent emission (ie figuring out the source).  Might they point the way for a future generation of researchers to improve on their methods? So, for example, if the coral is reflecting the light, it should be clear in a multi-spectral image or, perhaps even illuminate with NIR and look at reflectivity in this wave band.?

Author Response

Response to Reviewer#1-Second Comments

There are a few self-laudatory expressions that I would encourage them to remove.  In general, they don't really add content and are a bit self proclaiming.  For example:  "Given that virtually all of the Chl in corals is contained within the zooxanthellae, our NIR Chl fluorescence images are  among the best and most selective images of zooxanthellae distributions in coral insituever obtained".

This change was made exactly as suggested by the reviewer (but comparing our images to those from many other publications does not alter our “opinion” of the quality of our work).

The authors rightfully argued that the depth of the corals would preclude the NIR light from penetrating that far down. However, this new version does highlight that NIR emission due to CHL fluorescence could be reflected from skeletons and would then be part of the images. I'm not sure that we would say "mistaken' but that would be my comment.  This is the major improvement that adds content that warrants publication.

We appreciate this comment from the reviewer and believe our previous modifications to the text to note that our images are a combination of direct and reflected light from Chl have significantly improved the clarity of our work.

In general, it's always nice to point out some future directions with provisos on the limitations of the methods. So.. for example they do agree that (quote from them: "Thus, imaging both green and red fluorescence and simply separating the channels is fraught with complications, requiring more extensive characterization. All of the pixels, green, red and blue, respond to NIR light, albeit with non-identical efficiencies."  I think it would be nice to include such a statement in the text.  In addition, it seems like more advanced narrow band studies that looked at multiple narrow band reflections with concomittent imaging technologies (narrow band) could resolve the conundrum of definitely assigning reflected NIR vs inherent emission (ie figuring out the source).  Might they point the way for a future generation of researchers to improve on their methods? So, for example, if the coral is reflecting the light, it should be clear in a multi-spectral image or, perhaps even illuminate with NIR and look at reflectivity in this wave band.?

We have altered the final paragraph to two new paragraphs in the discussion incorporating these suggestions.  We are currently going in a few different directions. The first is the attempt to be able to remove reflected NIR light by image math 735fl - 735refl; although there are commercially available 10, 25 & 50 nm band pass filters near both the 680 and 735 peaks (Edmund Optics), doing the post-processing analysis will be time consuming.  Hyperspectral fluorescence imaging might be an alternative work around, albeit equipment limitations and increased cost will likely remove this option from citizen scientists (Chennu, Arjun, et al. "A diver-operated hyperspectral imaging and topographic surveying system for automated mapping of benthic habitats." Scientific reports 7.1 (2017): 7122.).  If corrected fluorescence and reflected NIR light is successful (above), the last development would be an attempt to quantify Chl fluorescence by a combination of reflectance subtraction and ratio imaging (735fl-735ref/700fl-700/ref: Gitelson, Anatoly A., Claus Buschmann, and Hartmut K. Lichtenthaler. "The chlorophyll fluorescence ratio F735/F700 as an accurate measure of the chlorophyll content in plants." Remote Sensing of Environment 69.3 (1999): 296-302).  Depending upon the promise of the quantitative work, an attempt to actually see if the Chl content during bleaching events in situ (cur ref 68).  Finally, if this type of analysis can then be performed by autonomous underwater vehicles (AUVs) (Mogstad, Aksel Alstad, Geir Johnsen, and Martin Ludvigsen. "Shallow-Water Habitat Mapping using Underwater Hyperspectral Imaging from an Unmanned Surface Vehicle: A Pilot Study." Remote Sensing 11.6 (2019): 685.), the methodology we introduce here could actually be capable of broad use in conservation efforts.  (We recognize that there is an extraordinary amount of work to be done in this vein, with numerous controls, but if this initial work can foster others to pursue this vein of research, this paper will have served a vital function).